# The Role of Cyclodextrins in COVID-19 Therapy—A Literature Review

**DOI:** 10.3390/ijms24032974

**Published:** 2023-02-03

**Authors:** Beatriz Almeida, Cátia Domingues, Filipa Mascarenhas-Melo, Inês Silva, Ivana Jarak, Francisco Veiga, Ana Figueiras

**Affiliations:** 1Laboratory of Drug Development and Technologies, Faculty of Pharmacy, University of Coimbra, 3000-548 Coimbra, Portugal; 2LAQV-REQUIMTE, Laboratory of Drug Development and Technologies, Faculty of Pharmacy, University of Coimbra, 3000-548 Coimbra, Portugal; 3Institute for Clinical and Biomedical Research (iCBR) Area of Environment Genetics and Oncobiology (CIMAGO), Faculty of Medicine, University of Coimbra, 3000-548 Coimbra, Portugal

**Keywords:** cyclodextrins, antiretroviral drugs, inclusion complexes, COVID-19, SARS-CoV-2

## Abstract

Coronavirus disease-19 (COVID-19) emerged in December 2019 and quickly spread, giving rise to a pandemic crisis. Therefore, it triggered tireless efforts to identify the mechanisms of the disease, how to prevent and treat it, and to limit and hamper its global dissemination. Considering the above, the search for prophylactic approaches has led to a revolution in the reglementary pharmaceutical pipeline, with the approval of vaccines against COVID-19 in an unprecedented way. Moreover, a drug repurposing scheme using regulatory-approved antiretroviral agents is also being pursued. However, their physicochemical characteristics or reported adverse events have sometimes limited their use. Hence, nanotechnology has been employed to potentially overcome some of these challenges, particularly cyclodextrins. Cyclodextrins are cyclic oligosaccharides that present hydrophobic cavities suitable for complexing several drugs. This review, besides presenting studies on the inclusion of antiviral drugs in cyclodextrins, aims to summarize some currently available prophylactic and therapeutic schemes against COVID-19, highlighting those that already make use of cyclodextrins for their complexation. In addition, some new therapeutic approaches are underscored, and the potential application of cyclodextrins to increase their promising application against COVID-19 will be addressed. This review describes the instances in which the use of cyclodextrins promotes increased bioavailability, antiviral action, and the solubility of the drugs under analysis. The potential use of cyclodextrins as an active ingredient is also covered. Finally, toxicity and regulatory issues as well as future perspectives regarding the use of cyclodextrins in COVID-19 therapy will be provided.

## 1. Introduction

Infectious diseases are responsible for millions of deaths yearly, and the associated economic costs for preventing and treating them are huge. For this reason, it is increasingly important to understand how these diseases evolve to reduce their impact on the socioeconomic landscape and health sector [1]. 

In 2019, a new coronavirus disease (COVID-19) caused by the SARS-CoV-2 virus emerged, quickly becoming a pandemic. This disease is transmitted from person to person by the inhalation of droplets released during coughing or sneezing. Although several body systems can be affected, the most impacted is the respiratory system, and symptoms can range from fever to pneumonia. While some patients are asymptomatic or only present mild symptoms, some develop severe symptoms with poor prognosis [2]. 

Since its beginning, many scientific resources have been devoted to better understanding the virus and finding the best treatment. Several therapies have been developed, including vaccines, antivirals, monoclonal antibodies, and others. Despite vaccines playing a pivotal role in preventing and containing the spread of the virus, antiviral agents have played a key role in treating the disease. However, drawbacks have limited their translation to clinics, namely low bioavailability, and in some cases, adverse events that have been reported.

The use of cyclodextrins (CDs) allows the formation of inclusion complexes (ICs) for the benefit of certain drugs, providing more safety and greater efficacy [1,3]. 

CDs are cyclic oligosaccharides composed of six or more glucose units connected by alpha-1,4 bonds. They have hydrophobic cavities with a hydrophilic exterior which gives them the ability to complex with several drugs, improving their solubility, stability, and bioavailability [3,4]. Due to these versatile properties, the global CDs market size is projected to increase from USD 260 million (2020) to more than USD 390 million (2027), with a compound annual growth rate (CARG) of 5.5 % [5]. 

The potential application of CDs in the encapsulation of antiviral drugs such as favipiravir (FPV), remdesivir (REM), dexamethasone (DEX), ivermectin (IVM), hydroxy(chloroquine) (HCQ), interferon-beta (IFN-β), lopinavir/ritonavir (LPV/RTV), oseltamavir (OTV), and fenofibrate have been explored with some promising results. This manuscript brings together the ongoing clinical trials with these drugs, at the time of this review, to provide more detailed information about the main specifications of their use, and to allow a clearer analysis of the feasibility of their incorporation into CDs. In addition, this work provides updated information about the new proposals for therapeutic approaches for the treatment of COVID-19, highlighting new candidate drugs for this purpose: bepridil, glycyrrhizin, plitidepsin, thapsigargin, and polyphenols. Experimental studies of these drugs with CDs are also mentioned as well as the ongoing clinical trials.

This review also aims to raise awareness of the importance of toxicological analysis of CDs, focusing on aspects such as the daily dose administered, the route of administration, and the type of cyclodextrin. These aspects, together with the information collected from clinical and experimental studies, are essential to make available conclusions on the viability of incorporating these drugs into CDs.

Therefore, this review aims to provide a comprehensive overview of the current therapeutic regimen against COVID-19 and the ongoing clinical trials, highlighting the potential use of CDs to overcome some therapeutic failures. Moreover, the potential application of CDs in new therapies was emphasized. Finally, the regulatory landscape of cyclodextrins will be covered.

## 2. COVID-19 Etiopathology

COVID-19 is an infectious disease declared as a pandemic on 11 March 2020 by the World Health Organization (WHO) [6]. Briefly, COVID-19 is caused by severe acute respiratory syndrome coronavirus 2 (SARS-CoV-2), which belongs to a diverse group of coronaviruses characterized by enveloped, single-stranded, positive sense ribonucleic acid (RNA) viruses with a long-range tropism, which gives them the ability to cause overwhelming diseases [7]. 

Viruses are known to enter the host cell with the help of receptors that mediate endocytosis. In the case of SARS-CoV-2, it has been reported that the spike protein is responsible for binding to the angiotensin-converting enzyme 2 (ACE-2) receptor on the host cell surface, which is the entry point for the virus. However, SARS-CoV-2 entry is not only dependent on the binding of the spike protein to the ACE-2 receptor, but it also requires the priming of the spike protein by the serine-2 transmembrane protease (TMPRSS2), which is crucial for the fusion of the virus to the host cell membrane. This synergy between the ACE-2 receptor and TMPRSS2 is necessary for the virus to enter into the host. The expression of TMPRSS2 is much higher than the ACE-2 receptor, suggesting that the latter is the limiting factor for SARS-CoV-2 during the early stage of infection [8,9]. 

Evidence has shown that the first target of the virus is the respiratory system. However, it can cause alterations in different body organs. Although some patients are asymptomatic or have mild to moderate symptoms, a percentage can develop severe illness [10]. The most common complications are acute respiratory failure syndrome (ARDS), septic shock, and sepsis. Risk factors such as age and comorbidities such as chronic diseases are related to severe illness and mortality [2]. 

Several symptoms have been associated with this disease, such as fever, cough, difficulty in breathing, headache, fatigue, sore throat, rhinorrhea, anorexia, myalgias, diarrhea, and in severe cases, pneumonia. The primary transmission mode is from person to person, through inhalation of the droplets released when coughing or sneezing. In general, symptomatic people are more contagious. However, transmission is also possible through fomites [2].

Moreover, it is valuable to mention that viruses are susceptible to mutations leading to the potential development of new variants. In response to the emergence of new SARS-CoV-2 variants, WHO has classified the variants according to the Greek alphabet (e.g., Alpha, Beta, Gamma, and Delta). 

The strains of interest present mutations on the spike protein, which often results in altered virus comportment and may lead to immune escape [2,7,11]. 

Therefore, comprehensive knowledge of the current prophylactic and treatment schemes for COVID-19 is of the utmost importance to preparing efforts to fight disease dissemination.

## 3. COVID-19 Prevention and Treatment Approaches

As the pandemic evolved, so did the search for potential prophylactic and therapeutic agents (Figure 1). 

Vaccines have undoubtedly conquered the research space to prevent SARS-CoV-2 because of their advantages in the prophylaxis of COVID-19. Besides that, some therapies have also been considered against COVID-19. Many of these therapies have emerged from drug repurposing. Repurposing of a drug consists of using an existing medicine with a new therapeutic purpose beyond its primary indications [7,12,13,14,15]. 

Table 1 summarizes the currently available therapies against COVID-19 and their main underlying action mechanisms. 

### 3.1. Vaccines

The spread of COVID-19 has mobilized research and development (R&D) efforts. Therefore, several approaches for vaccine development against COVID-19 have been tested, such as inactivated virus, live attenuated, recombinant protein, adenovirus vector, influenza virus vector, as well as mRNA and DNA vaccines. As a revolutionary innovation, mRNA vaccine technology has uniquely controlled the COVID-19 pandemic [19]. 

Briefly, mRNA vaccines are composed of a vehicle, particularly lipid nanoparticles, that enables the delivery of a nucleic acid molecule encoding the antigen of interest. In the case of SARS-CoV-2, the spike protein is delivered into the target cell in the human host, allowing the host cell to produce the target protein and express the antigen to elicit an immune response [19]. 

Currently, two mRNA-based vaccines are approved against COVID-19: Comirnaty (BNT162b2) and Spikevax (mRNA-1273) [20]. According to the literature, mRNA technology is desirable as it works as a template for protein translation and does not require bioreactors. It reduces the risk of bacterial contamination and makes scaling up less challenging. Moreover, mRNA vaccines reduce the risk of immunogenicity compared to other viral vector-based modalities. However, the dependency on cold-chain storage and transport may hamper their global applications [20]. 

### 3.2. Antiviral Drugs

Given the clinical picture presented by patients with SARS-CoV-2, another potential therapy is antiviral drugs such as remdesivir (REM), favipiravir (FPV), and lopinavir/ritonavir, which will be described in more detail later in the manuscript.

In brief, these drugs can inhibit the entry of the virus by targeting the type-II transmembrane serine protease (TMPRSS2) and the ACE-2 receptor. They may also interfere with endocytosis or with the action of RNA-dependent RNA polymerase (RdRp) and the SARS-CoV-2 3-chymotrypsin-like protease (3CLpro) through fusion inhibitors. 

Despite the promising prospects for this therapeutic class, some groups of antiviral drugs remain to be explored to treat COVID-19 [21,22]. 

### 3.3. Convalescent Plasma

The convalescent plasma of patients who have recovered from COVID-19 presents neutralizing antibodies in its constitution, which can fight infection by minimizing the inflammatory response [17]. The reduction in the inflammatory response may happen due to viremia suppression contributing to prophylaxis and recovery. 

The administration of passive antibodies may be an option to achieve rapid immunity [14].^.^ In theory, the administration of convalescent plasma should be completed at an early stage for superior efficacy [23]. However, its application continues to be controversial [24]. 

### 3.4. Monoclonal Antibodies

Monoclonal antibodies (mAbs) have effectively prevented and treated various viral infections [25]. Currently, potent neutralizing mAbs have been investigated against COVID-19, by targeting the receptor-binding domain (RBD) of the spike glycoprotein of SARS-CoV-2, blocking the binding between the S protein and the host receptor, ACE2 [26,27]. Moreover, other neutralizing mAbs can mediate viral activity by targeting nonblocking epitopes of the RBD or N-terminal domain (NTD) of the spike protein [28,29]. Some neutralizing antibodies studied against COVID-19 have been reviewed previously [30]. 

However, the emergence of new virus strains with mutations in the protein epitopes may hinder the application of these selective immunotherapies. 

Therefore, to overcome mutational virus escape, cocktail therapies aiming at administering antibodies targeting multiple epitopes on the spike protein have been investigated. Nevertheless, these treatment approaches may be challenging and considerably increase manufacturing costs [31]. 

Recently, the emergence of bispecific mAbs (bsAbs) has gained interest for the treatment of COVID-19 as one molecule can target two different antigen-binding sites [31]. However, the use of these approaches remains to be fully explored. 

Although applying mAb-based interventions against SARS-CoV-2 may require periodic updates due to the shifting antigenic landscape, the potential passive immunization in persons with a high risk of ineffective responses is a significant leap forward in the fight against viral evolution [32]. 

### 3.5. Interferons

Interferons (IFNs) induce the encoding of several proteins that can inhibit viral replication by decreasing cellular metabolism, interfering with the membrane formation necessary for virus replication, and inducing the release of cytokines that promote adaptive immunity. There are three families of IFNs, but only type I and type III are produced when the immune system detects the presence of viral nucleic acids. IFN-α belongs to type I, as well as INF-β, and fights coronaviruses by inhibiting virus replication [17,25]. 

According to Sodeifian et al. [33], it is paramount to establish the best time window to prescribe this type of treatment, as evidence has revealed that the administration of INF before the viral peak and the inflammatory phase of the illness could offer a highly protective effect. On the contrary, the administration of IFN during the inflammatory and severe phase of the disease may cause immunopathology and long-lasting harm for patients [33]. 

### 3.6. Corticosteroids

Corticosteroids are readily available agents extensively used as anti-inflammatory agents against respiratory infections. However, evidence has suggested that no clear benefits have been observed regarding their application in SARS and MERS patients. Therefore, their application in the initial phase of the COVID-19 pandemic was not recommended [34]. 

Later, due to preliminary data demonstrating lower mortality in patients with COVID-19 treated with corticosteroids, the use of corticosteroids for treating patients with severe or critical COVID-19 has been recommended [35]. 

Corticosteroids may play a pleiotropic role in different pathophysiological components in severe COVID-19 [36]. One of the studied drugs is dexamethasone (DEX), a synthetic glucocorticoid (detailed later) [17]. 

## 4. Cyclodextrins and Antiretroviral Agents 

CDs are versatile tools for drug delivery and produce effective inclusion complexes for antiretroviral therapies [3]. 

### 4.1. Definition and Structure

CDs are produced through the enzymatic degradation of starch from rice, corn, and potatoes by cyclodextrin glucanotransferases. These enzymes come from plants and microorganisms, usually *Bacillus* strains. CDs can be stored without detectable degradation for several years at room temperature [4]. 

CDs are cyclic oligosaccharides composed of six or more glucopyranoside (Glpc) units connected by alpha-1,4 bonds (Figure 2). 

Depending on the number of Glpc units, CDs can exist in various forms: α-CD (six units), β-CD (seven units), or γ-CD (eight units), as illustrated in Figure 2 [37]. CDs are also formed from nine or ten units, being classified as δ-CDs and ε-CDs, respectively. 

Larger CDs known as ζ-CD, are formed from eleven units, although they present poor complexation capability compared to naturally occurring CDs. 

The first three CDs listed are of natural origin and are crystalline and homogeneous substances. Some of their properties are listed in Table 2. Therefore, the most effective ones are α-CD, β-CD, and γ-CD, since when the number of units is more than eight, the CDs are more expensive and lose their complex ability [3,4]. 

CDs present a truncated cone-like shape, and the hydroxyl groups are exposed on the opposite edges of the cone. At the narrow end, there are the primary 6-hydroxy groups of the glucose molecules, and at the broad end there are the secondary 2- and 3-hydroxy groups. This unique geometry allows for its high-water solubility while maintaining the size of the hydrophobic cavities. This cavity allows the encapsulation of hydrophobic substances; therefore, they form inclusion complexes without modifying the structure and chemical properties of the host. Their cylindrical structure provides them with advantages such as being more resistant to enzymatic degradation and hydrolysis. Moreover, it confers improved complexing properties and a higher solubilizing potential than linear dextrins [3,4]. 

CDs have been used for a variety of functions (Figure 3). Indeed, their versatility makes them suitable to be applied as excipients in various fields, such as in the food industry, cosmetics, chemicals, catalysts, agriculture, and biotechnology. CDs have also been employed to increase drug solubility and bioavailability, work as stabilizer agents, and promote antiviral action, operating as antigen vectors, vaccine adjuvants, and antibody stabilizers [3]. 

The most relevant CD in the pharmaceutical industry is β-CD due to its cavity size, which is the most suitable for the most common drugs, its simple production, effective complexation, low cost, and ability to increase bioavailability and reduce toxicity. Although it has a lower aqueous solubility than α-CD and γ-CD, it constitutes about 90% of the CDs used. Other disadvantages of β-CD are the high affinity for cholesterol, which leads to nephrotoxicity due to the crystallization of β-CD-cholesterol complexes in the kidney, which are poorly soluble in water [4]. 

Chemically modified CDs are obtained from natural CDs through substitutions and the functionalization of the hydroxyl groups. The need to develop CD derivatives arose due to several factors. Among them is the limited solubility in water due to the intramolecular hydrogen bonds of the naturally occurring CDs. For this reason, chemically modified CDs have been demonstrated to improve and increase natural CDs’ physical and chemical properties and their inclusion capacity. 

The use of propylene oxide with α-CD, β-CD, and γ-CD leads to the hydroypropylated CDs derivatives; for example, hydroxypropylated-β-cyclodextrin (HP-β-CD) or hydroxypropylated-γ-cyclodextrin (HP-γ-CD). On the other hand, the application of methyl iodide drives the formation of randomly methylated CDs (e.g., randomly methylated β-cyclodextrin (RM-β-CD)). The employment of 4-butane sultone has also been reported to produce sulfobutylether CDs (e.g., sulfobutylether β-cyclodextrin (SBE-β-CD)) [4]. 

The formation of ICs (Figure 4) can be explained based on the structure and characteristics of the CDs. The interaction between the CD cavity and the hydrophobic part of the drug results in the formation of the drug–cyclodextrin complex. The organic compounds are incorporated into the CD cavity and, consequently, the drug’s physical, biological, and chemical alterations are made. In addition to improving the physicochemical stability of the therapeutic molecule, the formation of these complexes also interferes with the release kinetics, and pharmacodynamic and pharmacokinetic properties [4]. 

The complexing ability of CDs has allowed insoluble drugs to improve their bioavailability by improving release profile, dissolution rate, chemical stability, and absorption efficiency. All these factors increase the oral bioavailability of the drug and enhance its biological activity [4]. 

Despite the many functions of CDs, the focus of this work is on their application in antiviral drug delivery. CDs interact with the various active pharmaceutical ingredients (APIs) used to treat viral diseases, either by forming ICs or by using the excess amounts to create products with increased solubility and/or activity [3,39]. 

Drugs with antiviral activity are aimed at fighting viral infections. However, their physicochemical properties, such as solubility, stability, and permeability, may hamper their clinical translation. CDs have been used as excipients to protect and slowly release the active ingredient to optimize the bioavailability and distribution of these drugs. 

The use of CDs as antiviral drug delivery systems has been properly explored [39]. The next chapter will address some drugs that are currently being studied against COVID-19, and the application of CDs to overcome some challenges will be underlined. 

### 4.2. Cyclodextrins and Anti-SARS-COVID-19 Molecules 

Despite the intense focus on vaccine development, searching for drugs that alleviate the symptoms of SARS-CoV-2 has not been abandoned. To this end, molecules already known were used in various studies to determine whether they could directly or indirectly inhibit the spread of the virus (Figure 5). 

Some of these treatments have revealed certain limitations. Therefore, the use of CDs as host molecules for anti-COVID-19 drugs has been exploited. Supramolecular interaction studies were performed with the corresponding α-, β-, and γ-CDs and showed that the stoichiometry of the complexes formed was 1:1. Some of the proposals announced were the use of SBE-β-CD as a solubilizer of REM, HP-β-CD for the lopinavir/ritonavir combination, and the use of β-CD as a flavor modifier for oseltamivir [3,39]. 

Moreover, the combination of ritonavir with other antiviral agents, such as nirmatrevil (PF-07321332), has been addressed in clinical trials and has shown clinical efficacy in reducing hospitalization by 80% [40]. PF-07321332 is a reversible covalent inhibitor of the M^pro^ related to SARS-CoV-2 that binds to the catalytic cysteine (Cys145), interrupting the viral replication cycle [41,42]. According to EMA/783153/2021 (16 December 2021), PF-07321332 belongs to the BCS II/IV, presenting low solubility with permeability to be clarified. The use of CDs for oral administration is not fully exploited. However, the application of 2-hydroxypropyl-β-CD in the formulation of PF-07321332 for IV administration in monkeys has been tested [41]. 

#### 4.2.1. Favipiravir 

Favipiravir (FPV) (Figure 5) is a prodrug and a purine base analog that is metabolically activated by phosphoribosylation to yield the activated metabolite FPV-ribofuranosyl-50-triphosphate (T-705RTP). 

The primary mechanism of action is based on the binding and inhibition of the *RNA-dependent RNA polymerase* (RdRp), preventing transcription and replication of viral genomic RNA. There are several proposals to explain the interaction between FPV and RdRp, one of which is that T-705RTP inhibits viral RNA synthesis by terminating the nascent viral RNA chain [43,44]. 

The pharmacokinetics of FPV are complex and nonlinear, depending on time, dose, and weight. Bioavailability is almost complete at 97.6%; maximum concentration is 51.5 µg/mL; peak concentration occurs at 1 h, and half-life ranges from 4.8 to 5.6 h. In a phase 3 clinical trial, FPV has been reported to be metabolized and inhibited by an aldehyde oxidase (AO). Therefore, an appropriate starting dose is required to maintain adequate blood levels. Based on single-dose toxicity studies, the oral and intravenous lethal dose of FPV in mice is approximately >2000 mg/kg. In rats, the oral lethal dose is >2000 mg/kg, while in dogs and monkeys, it is >1000 mg/kg [43,44]. 

Clinical trials that are currently recruiting using FVP are summarized in Table 3.

Reported adverse effects of FVP in men were well tolerated in clinical trials but are associated with increased uric acid and should be used with caution in patients with a history of gout or hyperuricemia. Other adverse effects may include diarrhea, increased transaminases, and decreased neutrophil counts [44]. 

The disadvantages of FPV include that it is poorly soluble in physiologically buffered saline (PBS) (≈0.01%) and its effective dose is relatively high (600–1600 mg/day). A 1:1 stoichiometry complexation of β-CD and FPV would require an oral intake of approximately 4.3–13.5 g/day of β-CD, which significantly exceeds the European Medicines Agency (EMA) recommendation (0.5–1 g/day). Other CDs are well tolerated in higher amounts, but their molecular weight is higher than that of β-CD. 

Some of the literature is already available on the behavior of the FPV/CD complex. However, its structure suggests that the formation of this complex would maintain only constant low stability and marginally improve its water solubility [3,44]. 

#### 4.2.2. Remdesivir 

Remdesivir (REM) (Figure 5) has potent viral activity against mRNA viruses, and after the pandemic began, it has been tested against SARS-CoV-2 with positive results in some clinical trials [45]. 

In April 2020, REM was approved by the EMA to be used in critically ill patients with COVID-19. REM was also the first antiviral approved by the Food and Drug Administration (FDA) for the treatment of COVID-19 [45,46,47]. 

REM is a prodrug transformed metabolically by intracellular esterases and kinases for a pharmacologically active nucleoside, a triphosphate analog (GS443902). This active metabolite competes with adenosine phosphate and acts as an inhibitor of RdRp, thereby inhibiting viral replication. 

The use of REM has also been presented in clinical trials, as reviewed in Table 4.

Because of its hepatic first-pass metabolism, oral bioavailability is low. Therefore, the alternative is intravenous infusion as a parenteral solution. However, its low water solubility (0.028 mg/mL) at neutral or slightly acidic pH hampers its parenteral administration. 

To overcome this problem, excipients with solubilizing capabilities, such as surfactants, polymers, co-solvents, and SBE-β-CD can be used [46,47,48]. 

In the case of REM, satisfactory solubility was achieved using SBE-β-CD, which now has values between 7.6 and 9.7 mg/mL. Although SBE-β-CD has a much poorer complexing capacity than HP-β-CD, its anionic properties can increase the solubility of ionizable molecules at a tolerable pH. It should also be added that although adverse effects on the kidney have been reported from SBE-β-CD, they are less than those observed for HP-β-CD, albeit the drug/CD ratio is worse [3,39,47]. 

Vámai et al. [46] have studied the molecular interactions of REM in the cavity of CDs and concluded that the ethylbutyl moiety in β-CDs performs a crucial role in the complexation of REM into the β-CD cavity (Figure 6). While in the case of γ-CDs, the phenoxy moiety is the main on responsible for the REM complexation in the γ-CD cavity [46]. 

Moreover, the complexation of REM into SBE-β-CD is preferably at a low pH when REM is protonated, which increases solubility. Even though it was expected that the strength of the complex would decrease with an increasing pH, it was shown that the drug remains encapsulated by the addition of NaOH, so that it can be administered in the human body at a neutral pH. The influence of the number of CD substitutions on the affinity constant in the formation of inclusion complexes was also studied, and the conclusions were that the lower the number of substitutions, the lower the affinity constant, i.e., inclusion complexes with 6SBE and 7SBE have a higher affinity than those with 5SBE [48,49]. 

When using CDs to increase the solubility and stability of a molecule, it is necessary to quantify the affinity between the two compounds and to know the structure that results from their complexation. A very high-affinity constant is not optimal because the molecule may not be released. In contrast, a low affinity constant will cause solubility to increase and protection from degradation to become inadequate [48]. Some studies have reported the application of REM combined with CDs in clinical trials, which will be discussed further [50,51]. 

#### 4.2.3. Dexamethasone 

DEX (Figure 5) belongs to the corticosteroid class and has anti-inflammatory and immunosuppressive properties. Its indications are the treatment of asthma, allergic reactions, arthritis, and other autoimmune diseases. After demonstrating its efficacy in reducing mortality in critically ill COVID-19 patients, it was included in the National Institutes of Health (NIH) recommended list, although its viral activity is still under investigation. The administration of low doses of DEX reduces the risk of death by up to one-third in patients requiring ventilators, and by one-fifth in patients requiring oxygen without invasive ventilation. It is ineffective in non-severe cases but has not caused significant adverse effects. However, DEX can induce hormonal changes, fluid retention, weight gain, anxiety, and sleep disturbances [3,52]. 

The mechanism of this drug is to suppress the immune system by blocking two pathways of inflammation, vasodilation, and immune cell migration. 

DEX penetrates the host membrane and binds to the glucocorticoid receptors present in the cytoplasm. This binding leads to a series of immune cell responses resulting in the suppression of pro-inflammatory cytokines such as interleukin 1,2,6,8 (IL-1, IL-2, IL-6, IL-8), tumor necrosis factor (TNF), and interferon-γ (IFN-γ). It also increases the expression of interleukin-10 (IL-10), an anti-inflammatory cytokine, and inhibits neutrophil adhesion to endothelial cells, preventing the release of lysosomal enzymes and chemotaxis at the site of inflammation. Moreover, it also inhibits macrophage activation, which is important for the cytokine storm in SARS-CoV-2-infected patients. Other advantages of this drug include its long duration of action, which allows for a once-daily administration and lowers costs, and the fact that it is already on the market [52]. 

The use of DEX is being addressed in some recruiting clinical trials for the treatment of COVID-19 (Table 5).

DEX is a hydrophobic drug that hampers achieving therapeutic concentrations when administered orally. Hence, the entrapment of DEX in CDs has been exploited to increase its water solubility and bioavailability while reducing adverse effects. A study showed that the inclusion of dexamethasone in β-, γ-, and HP-β-CD could be an interesting approach for drug delivery. The result showed that the DEX/β-CD complex is a suitable strategy to overcome the problems of low solubility and improve bioavailability, which also enhances performance in the treatment of COVID-19 [53]. 

The encapsulation mode of DEX into the cavity of β-CD as an inhibitor of the COVID-19 main protease has been investigated using density functional theory with the recent dispersion corrections D4 and molecular docking (Figure 7) [53]. 

#### 4.2.4. Ivermectin

The FDA approved Ivermectin (IVM) (Figure 5) is a broad-spectrum antiparasitic and it has shown antiviral activity against several viruses in vitro. In addition to its viral activity, it has also demonstrated anti-inflammatory activity [54]. 

IVM was identified as an inhibitor of the interaction between the human immunodeficiency virus (HIV-1) and the importin (IMP) α/β1 heterodimer, responsible for the HIV-I integrase nuclear import and replication. Moreover, IVM has also proved to inhibit the host nuclear import and viral proteins [55,56]. 

IVM has been reported for the prophylaxis and treatment of COVID-19 [57]. Recruiting clinical trials using IVM for the treatment of COVID-19 are summarized in Table 6. 

However, the application of IVM in COVID-19 remains controversial [58]. 

One proposed solution to overcome the limitations of IVM, such as poor solubility, bioavailability, and neurotoxicity, is to develop a formulation aiming at a local action. For that, HP-β-CD has been employed to formulate an inhaled dry powder of IVM [59]. The use of HP-β-CD has increased the aqueous solubility of IVM 127-fold. The inhalation of IVM-HP-β-CD at 0.05 and 0.1 mg/kg revealed a safety profile in rats [59]. 

#### 4.2.5. Interferon-Beta

IFN-β is a cytokine produced in mammalian cells and IFN-β 1b is produced in modified *Escherichia coli* cells. When the human organism is exposed to chemical or biological stimulation, the immune system initiates its production. It exhibits antiviral and anti-inflammatory properties and activates the immune system. Its antiviral action is based on inhibiting viral replication through interaction with Toll-like receptors (TLRs) [60]. 

In cell culture and animal experiments, interferons have been shown to inhibit coronavirus replication. In addition, studies have demonstrated the efficacy of IFN-β against SARS-CoV-2 in vitro and strong activity to reduce the replication of MERS-CoV. 

Therefore, considering this information, IFN-β was redirected to treat COVID-19. Observations made after the initiation of its use were a significant reduction in length of a hospital stay, mortality, intensive care unit (ICU) admission, and intubation rate [60]. 

However, IFN-β use is limited by its short half-life, side effects, route of administration, and limited access to the central nervous system (CNS). The fact that IFN-β must be administered in high doses may increase the occurrence of adverse effects and consequently affect patient compliance by decreasing efficacy. If the route of administration is subcutaneous or intramuscular, problems with the production of neutralizing anti-IFN-β antibodies (NAB) may occur, reducing bioavailability and efficacy. Currently, two clinical trials are recruiting using INF-β for inhalation (NCT04469491) and intravenous bolus injection (NCT02735707) against COVID-19. 

The systemic administration of IFN-β may cause inflammation, erythema, or even necrosis at the administration site [61]. 

Hence, the use of CDs may be the desired solution to overcome these problems. For example, methylated CDs may positively affect the secretion of IFN-β through cholesterol-RM-β-CD interactions in cell membranes. However, their use in the treatment of SARS-CoV-2 is still unclear [3]. There is also a study linking CDs to IFN-β in the context of multiple sclerosis treatment. This study reports a formulation consisting of chitosan and SBE-β-CD loaded with IFN-β to deliver cytokines intranasally to the CNS. However, further research is needed to relate the use of CDs and IFN-β to the treatment of COVID-19 [61]. 

#### 4.2.6. Lopinavir/Ritonavir

Lopinavir (LPV) (Figure 5) is a drug used to treat infections caused by the HIV-1 virus. LPV belongs to the biopharmaceutics classification system (BSC) class IV, meaning that it has low permeability and solubility in aqueous media, namely 2.27 µg/mL in water and 2.93 ± 0.08 µg/mL in PBS at pH 7. In addition to these properties affecting absorption in the gastrointestinal tract, its bioavailability is also reduced by cytochrome P-450 metabolism and P-glycoprotein efflux transport [45,62]. 

Ritonavir (RTV) belongs to the class of protease inhibitor drugs. It is particularly important in improving the pharmacokinetics of LPV against HIV-1 protease [63]. Considering this, the combination of LPV with RTV has been addressed in COVID-19 (Table 7) [64]. 

Moreover, to enhance the bioavailability of LPV, the application of CDs has been addressed [3,45]. 

In a study conducted by Adeoye et al. [62], it was demonstrated that CDs improved the solubility of LPV with more pronounced effects for the newly synthesized form of HP-γ-CD with a high number of substitutions ((HP)-17-γ-CD). Later on, the same group crosslinked pyromellitic dianhydride (PMDA) with two CD derivatives (methyl-β-CD-MβCD and (2-hydroxy)propyl-β-CD-HPβCD) and observed an increase in the solubilization of LPV by 12–14 fold and antiviral activity against HIV-1, particularly with pMβCD [65]. 

#### 4.2.7. Oseltamivir

Oseltamivir (OTV) (Figure 5) is an antiviral drug designed to influenza viruses (A and B) in patients with a high risk of complications. 

Its antiviral activity is based on blocking the viral activity of the enzyme neuraminidase, which is located on the surface of the virus, thus preventing viral replication. Therefore, the efficacy of OTV has been evaluated in several clinical trials to treat SARS-CoV-2 virus infections [3,66]. 

Based on a search performed on clinicaltrials.gov (access date 28 November 2022, range from 1 January 2022 to 31 December 2022), two clinical trials are using OVT for COVID-19 in a recruiting stage addressed at the adult population. One is a pharmacovigilance study (Phase 4, NCT04973462) and the other is an interventional study (Phase 3, NCT02735707) aiming the administration of OTV twice daily for five days across a total of 10 days via enteral administration. 

According to the EMA report, EMA/CHMP/315246/2014, OTV is a compound with limited absorption and scarce data on its solubility, which makes its BCS classification difficult. However, taking into consideration the bioequivalence requirements, if the applicant generates solubility data and classifies the drug according to the BCS criteria as highly soluble, oseltamivir could be classified as a BCS class III drug, and a BCS biowaiver could be applicable. 

In previous studies, this molecule has demonstrated moderate aqueous solubility (≈0.2%) and satisfactory bioavailability (>80%) [3]. 

Oral suspensions of oseltamivir phosphate are dispensed orally in capsules and suspension. However, an oral suspension is preferable for pediatric administration. Therefore, OVT is prepared as a powder for reconstituted suspension, which degrades in a few days [67]. Hence, HP-β-CD was used to formulate ICs with OTV to improve solubility and stability, especially in formulations where OTV is suspended and needs to be reconstituted. This formulation showed an extended shelf life of more than 90 days without affecting the properties of the active ingredient [67]. 

In a recent study, Rajamohan et al. tested methylated-β-Cyclodextrin (M-CD) and sulfated-β-Cyclodextrin (S-CD) as inclusion complexes for OTV. Their results showed that OVT was successfully complexed into the cavity of both modified M-CD and S-CDs (Figure 8(A1,A2), respectively), offering increased antiviral activities with more pronounced effects when using S-CD (Figure 8B) [66]. 

#### 4.2.8. Fenofibrate

Fenofibrate (Figure 5) is a drug that belongs to the class of fibrates and is used to treat lipid metabolic disorders such as hypercholesterolemia, hypertriglyceridemia, and dyslipidemia. 

The mechanism of action of the SARS-CoV-2 virus suggests the upregulation of genes related to lipogenesis and the process of cholesterol synthesis in bronchial epithelial cells. SARS-CoV-2 infection appears to cause lipid deposition in the lung, which may control the severity of the disease. Hence, it has been suggested that fenofibrate may be a potential agent against COVID-19 by reducing the replication of SARS-CoV-2 in lung cells [3,68,69]. However, based on the clinical trial NCT04517396, fenofibrate has not led to decreased COVID-19 severity [70]. 

Currently, a phase 3 clinical trial (NCT04661930) is recruiting the adult population to administer fenofibrate p.o. once daily for 10 days in patients with COVID-19 (clinicaltrials.gov, access date 28 November 2022, ranging from 1 January 2022 to 31 December 2022). 

Fenofibrate belongs to BCS class II, meaning it is poorly soluble, which affects its bioavailability and permeability, although the latter is high. Its bioavailability profile ranges from 60% to 81%. 

Therefore, to improve the physicochemical properties of fenofibrate, several studies have been performed using CDs to form ICs, such as HP-β-CD, which showed the improved absorption and solubilization of fenofibrate in an aqueous medium [3,68,71]. 

#### 4.2.9. Cetylpyridinium Chloride

Cetylpyridinium chloride (CPC) (Figure 5) is a quaternary ammonium compound commonly used as an antimicrobial agent in oral hygiene products. The main components of mouthwashes have an antiseptic action and help to reduce the spread of microorganisms [72,73]. 

Recently, CPC has been shown to exhibit antiviral activity against the SARS-CoV-2 virus. The mechanism of action is related to the disruption of the viral lipid membrane. This disruption of the lipid membrane will interfere with the ability of the virus to enter cells, which may reduce the viral burden in saliva and the risk of transmission [73,74]. 

Currently, a clinical trial (NCT05178173) is ongoing, aiming to inactivate the SARS-CoV-2 virus in the saliva of COVID-19-positive patients using antiseptic mouth rinses (Volume of 20 mL) with 0.075% CPC. 

The use of CDs to enclose CPC has not been fully explored. However, an expired patent (CA1314225C, Canada) mentions the application of CDs to enclose CPC.

## 5. New Candidates for COVID-19 Treatment

COVID-19 has gained huge attention due to its socioeconomic impact and health repercussions. Therefore, the scientific community has combined its efforts to identify new treatment approaches for COVID-19. Some of them are summarized in Figure 9.

### 5.1. Bepridil

Bepridil (Figure 9), a calcium channel blocker with significant antianginal activity, was reported to be potent against SARS-CoV-2 in vitro [76]. The antiviral analysis of bepridil indicated that it has low micromolar EC50 values in inhibiting SARS-CoV-2 in two highly permissive mammalian cell lines, Vero E6 and A549/ACE2 cells [76]. The structure moiety N-phenyl-N-benzylamine may be responsible for the structure−activity relationship with the main protease and the potent effect against SARS-CoV-2 [76]. 

Bepridil has been reported as a class 1 according to the Biopharmaceutics Drug Disposition Classification System (BDDCS) [77], presenting high solubility and permeability. Therefore, according to the FDA, it can undergo biowaiver regulation. Moreover, due to its physicochemical properties, its complexation into CDs has not been reported, to the best of our knowledge.

### 5.2. Glycyrrhizin

Glycyrrhizic acid (GlyA) (Figure 9) is the major triterpene glycoside contained in licorice root (Figure 9) [78]. 

GlyA has been accepted as a treatment for chronic viral hepatitis C for over 20 years in Japan. Due to the availability of human safety data from GlyA intravenous administration in treating hepatitis C, GlyA has been proposed for COVID-19-infected patients using dose escalation studies under a compassionate use exception [79]. 

Interestingly, the use of GlyA nanoparticles against COVID-19 has been recently reported [80]. The formation of ICs of GlyA with CDs has revealed that the binding affinity of GlyA to γ-CD is about 300 times higher than that to β-CD [81]. The conjugation of GlyA complexed to β-CD has demonstrated beneficial results for treating influenza virus agents (Figure 10) [82]. However, the participation of this combination addressing COVID-19 treatment has not been exploited yet, to the best of our knowledge.

### 5.3. Plitidepsin

Plitidepsin (Figure 9) is a marine cyclic depsipeptide extracted from the ascidian *Aplidium albicans*. This compound has been actively studied due to its anticancer properties [83]. Indeed, in 2003, plitidepsin received orphan designation by the EMA (EU/3/03/151). Lately, plitidepsin has been repurposed for treating multiple myeloma because it targets cofactor Eukaryotic translation elongation factor 1 alpha (eEF1A) [84].

More recently, with the emergence of COVID-19, plitidepsin has been pointed out as a possible antiviral candidate against SARS-CoV-2 [84,85]. In fact, plitidepsin has shown antiviral activity against SARS-CoV-2 by inhibiting the activity of eEF1A. eEF1A participates in mRNA translation in RNA virus replication, being involved in the enzymatic delivery of aminoacyl tRNAs to the ribosome and the aminoacylation-dependent tRNA export pathway [86]. Plitidepsin can also inhibit the translation of the open reading frames (ORF) 1a and 1b, reducing the production of polyproteins (PP) and decreasing the amount of RNA-dependent RNA polymerase. Moreover, this drug is also responsible for inhibiting the translation of different subgenomic mRNAs, leading to a deficient production of viral structures and accessory proteins [86]. 

Preclinical data have revealed that plitidepsin presents potent antiviral effects against SARS-CoV-2, namely in infected Vero E6 and hACE2-293T cells, by reducing the expression of the viral structural protein N [85]. Moreover, plitidepsin underwent interventional clinical trials for COVID-19. The proof-of-concept phase 1 clinical trial, APLICOV-PC (NCT04382066), demonstrated the safety profile of plitidepsin. However, there were some limitations, namely the limited number of participants [87]. A phase 3 clinical trial, NEPTUNO (NCT04784559), is recruiting for hospitalized patients with COVID-19 of moderate severity. The therapeutic protocol includes the treatment groups that receive 1.5 or 2.5 mg/day of plitidepsin and DEX by intravenous administration and the control arms that only receive DEX [87]. 

Due to its hydrophobicity, with a LogP > 5, plitidepsin is nearly insoluble in aqueous media requiring an adjuvant to allow intravenous administration. Therefore, plitidepsin has been formulated using Cremophor® (CRE) and Tween 80 to increase drug solubility, although hypersensitive reactions have been reported. Taking it into consideration, other strategies have been proposed, namely the use of block copolymers, such as poly(ethylene glycol)-block-poly(γ-benzyl-L-glutamate) (PEG-b-PBLG) copolymer and poly(trimethylene carbonate)-block-poly(glutamic acid) (PTMC-b-PGA) [88]. To the best of our knowledge, the use of CDs to complex plitidepsin has not yet been addressed. 

### 5.4. Thapsigargin

Thapsigargin (TG) (Figure 9) is a sesquiterpene lactone found in the roots and fruits of the *Thapsia* L. species. TG works as a non-competitive inhibitor of the sarcoplasmic/endoplasmic reticulum Ca^2+^ ATPase pump (SERCA) with potential applications in anticancer therapy [89]. 

Recently, TG has been proven to present potent antiviral properties against influenza A virus replication by the ER stress unfolded protein response (UPR) [90]. More recently, TG has been proposed to be an acid-stable inhibitor of SARS-CoV-2 [91] and other respiratory viruses, efficient in separate infections as well as in co-infections. The ER stress response seems to be the main underlying antiviral action mechanism, although it remains to be fully explored [92]. 

TG encapsulation into poly(lactic-co-glycolic acid) (PLGA) nanoparticles for a target release has been exploited [93,94]. As far as we know, no study has addressed the application of CDs to carrier TG. However, an interesting experiment reports the modulation of TG store-dependent Ca^2+^ entry in macrophages by methyl-β-CD [95]. 

### 5.5. Polyphenols

Polyphenols are secondary plant metabolites that protect them against diseases, infections, and damage [96]. Polyphenols include, but are not limited to, phenolic acids, coumarins, flavonoids, stilbenes, and lignans (Figure 9) [75]. 

These molecules have been reported to present a range of health benefits [97,98], including in treating infectious diseases [99,100,101]. Therefore, the use of polyphenols as SARS-CoV-2 antiviral agents has been explored. For example, an in silico study testing green tea polyphenols, e.g., epigallocatechin gallate (EGCG), epicatechin gallate, and gallocatechin-3-gallate against COVID-19 have revealed that these three components can strongly interact with the catalytic residues of the SARS-CoV-2 main protease (M^pro^), constituting potential drug candidates for COVID-19 treatment [102]. Similarly, Ghosh et al. [103], using docking and molecular dynamics simulation approaches, have shown that the six polyphenols present in *Broussonetia papyrifera* can inhibit the catalytic activity of M^pro^. Therefore, due to the promising applications of polyphenols in treating COVID-19, it is of interest to expand the research to other registered polyphenols [104]. Based on this, Wu et al. [105] have conducted a large virtual screening for more than 400 polyphenols with the potential to bind to SARS-CoV-2 M^pro^ or papain-like protease (PL^pro^), which are central proteases to the viral life cycle. Their results revealed that several polyphenols, such as petunidin 3-O-(6″-p-coumaroyl-glucoside), present promising binding interactions with SARS-CoV-2 M^pro^ and PL^pro^ [105]. 

The translation of polyphenols from the bench to the bedside has already occurred for the treatment of COVID-19. Actually, resveratrol has been studied in a phase 2 clinical trial (NCT04400890) to evaluate its safety and explore its effectiveness for COVID-19. Moreover, the use of *Caesalpinia spinosa* extract is also being studied in patients with symptomatic COVID-19 (NCT04410510). Moreover, quadrate therapy of chicoric acid, 13-Cis retinoic acid (aerosolized), minocycline, and vitamin D has been explored for patients with multidrug-resistant COVID-19 (NCT05077813). Interestingly, glucoside- and rutinoside-rich crude has been investigated for vaccine-adverse reactions (NCT05387252).

Despite the promising applications of polyphenols in the treatment of COVID-19, they are susceptible to the negative impact of light, oxygen, and pH, which may hamper their extraction process and applications [106]. Therefore, the encapsulation of polyphenols into nanocarriers has been reported to circumvent these limitations [106]. 

Indeed, the use of CDs to complex polyphenols has already been reported with encouraging repercussions [107,108,109,110,111,112]. Moreover, taking advantage of sustainable green chemistry, CDs have been demonstrated to be promising contributors to the extraction of polyphenols [113,114,115]. 

## 6. Regulatory Issues and Toxicity

CDs have been used for a variety of purposes globally. Therefore, the need to regulate their applications has been promoted in Western countries. 

In Japan, natural CDs are presented in the Japanese Pharmacopoeia and are considered food additives [116]. 

Furthermore, the JECFA, an international expert committee composed of members of the Food and Agriculture Organization of the United Nations (FAO) and the World Health Organization (WHO) have worked on the regulation of native CDs in food and additives, being the pharmaceutical application of CDs under the responsibility of the EMA or the FDA [117]. 

From 2000 to 2004, the use of native CDs as food additives was declared “Generally Recognized As Safe” (GRAS) by the FDA [117]. Moreover, the chemically modified SBE-β-CD and HP-β-CD were listed as inactive pharmaceutical ingredients and can be used in oral formulations [118]. Due to their physicochemical properties, namely their high molecular weight and hydrophilic nature, low octanol-water partition coefficients, and the presence of several hydrogen bond donors and acceptors, CDs do not promptly permeate biological membranes through passive diffusion [116]. The oral availability of CDs is minimal, without significant absorption in the gastrointestinal tract. The total daily dose for α-CDs can reach 6000 mg and for γ-CDs 10,000 mg [118]. However, the acceptable daily intake of β-CD is restricted to 5 mg/kg [118]. Moreover, at high doses greater than 1000 mg/kg/day, the oral intake of CDs leads to reversible diarrhea and cecal enlargement (EMA/CHMP/495747/2013) [119]. CDs are poorly absorbed via mucosal membranes. However, at high doses, CDs can increase drug permeability by direct action on mucosal membranes, enhancing drug absorption and/or bioavailability by topical administration. These may be due to the solubilization of membrane components with mild and reversible perturbations on the cell membrane compared to surfactants that generally induce irreversible membrane damage [118]. For instance, nasal and pulmonary formulations containing 10% HP-β-CD, RM-β-CD, or less than 1.5% of β-CD have not shown tissue damage [119]. The use of CDs in rectal products has also been addressed, and the results revealed that in humans, the use of 230 mg of β-CD in suppositories did not induce irritation in rectal mucosa. In rabbits, the use of 12% of HP-β-CD did not cause rectal mucosal irritation. Alpha-CD can damage the epithelial cell layer [118], but no rectal product on the market reports its use [119]. Applying absorption-promoting agents can enhance the dermal absorption of CDs. The use of DM-β-CDs advantageously sustains a scent for a prolonged time compared to the use of surfactants. Alpha-, β-, and γ-CDs are safe for dermal applications at up to 0.1% [118]. CDs used in eye formulations have been reported to increase drug penetration and are not toxic or irritant for the eye of the rabbits when presented in a solution up to 10% (SBE-β-CD) or 12.5% HP-β-CD [119]. The parenteral administration of CDs has been reported to be favorable, but some safety and toxicity considerations should be addressed [120]. In fact, β-CDs have demonstrated pronounced hemolytic activity compared to α- and γ-CDs [116]. However, the use of α-CD, β-CD, and ME-β-CD has not been recommended for IV administration as they present nephrotoxicity at relatively low doses. The list of CDs not recommended for parenteral administration also includes RM-β-CD, which may only be used in topical formulations because of its high hemolytic activity and nephrotoxicity. On the other hand, the parenteral administration of HP-β-CD and SBE-β-CD have been considered safe at a concentration of ca. 250 mg/kg/day in humans older than two years when given for 21 days or six months, respectively [119]. 

Most of the regulators agree that CDs are excipients and not integral to the drug. However, this topic can be division- and product-specific and further studies to fully assess the toxicity associated with CDs are required [117,119,121]. 

## 7. Clinical Trials

Three clinical studies have stated the application of CDs in COVID-19 (Table 8).

Briefly, the CTRI/2021/05/033744 clinical trial aims to address the lack of information on the pharmacokinetics of REM and its vehicle SBE-β-CD in patients with renal disease, taking into account that both are excreted through the kidneys. Inclusion criteria include people between the ages of 18 and 90 of either sex with severe COVID-19 and renal disease who have an indication for treatment with REM [50]. 

The following clinical study, EUCTR2020-003486-19-GB, aims to compare the efficacy of Sulforadex or Sulforaphane/α-Cyclodextrin complex (SFX-01) versus placebo in treating patients with a suspected COVID-19 respiratory tract infection. Moreover, it also intends to evaluate the safety of SFX-01 and explore its underlying action mechanisms [122]. 

The last study, EUCTR2020-001803-17-GB, plans to evaluate the safety, tolerability, pharmacokinetics, and efficacy of REM by determining its antiviral activity and exposure to SBE-β-CD in patients up to 18 years with laboratory-confirmed COVID-19 [51]. 

## 8. Final Remarks and Future Perspectives

Despite the rapid development of vaccines and therapies that helped control COVID-19, an open path remains to eradicate the disease [123]. In this regard, this review article was intended to explore the potential of using cyclodextrins (CDs) as drug delivery systems for the treatment of viral infections, specifically for COVID-19. Resulting from the formation of inclusion complexes (ICs) between CDs and antiviral drugs, which improve the physicochemical properties of the drugs, CDs can improve the solubility, stability, and absorption of antiviral drugs as well as increase bioactivity and reduce toxicity. These complexes protect drugs from degradation, increase their solubility, and interfere with drug pharmacokinetics, improving their bioavailability and biological activity. In addition, the use of CDs can also allow for oral, inhalation, or topical drug administration, which is useful for avoiding side effects or administration problems associated with direct systemic administration. Chemical modification of CDs can also improve their inclusion properties and capacity. Considering the information gathered, it appears that β-CD is the most suitable and effective for encapsulating antiviral molecules due to its ability to increase the solubility, stability, and absorption of the molecules as well as reduce toxicity. In addition, β-CD is the most common form of cyclodextrin used due to its simple production, complexation efficiency, and low cost. However, it is important to note that other forms of cyclodextrins, such as α-CD and γ-CD, have also been studied and used to encapsulate antiviral molecules. Several anti-SARS-COVID-19 molecules are presented, including remdesivir, dexamethasone, ivermectin, interferon-beta, lopinavir/ritonavir, oseltamivir, fenofibrate, and cetylpyridinium chloride, but also new candidates such as bepridil, glycyrrhizin, plitidepsin, thapsigargin, and polyphenols which have been studied in combination with CDs to increase their efficacy in treating COVID-19. Although there are promising pre-clinical reports on CDs as drug nanocarriers for the treatment of COVID-19, their applications in clinical trials remain scarce. Some of the clinical trials mentioned are still in the early stages and have not yet been completed. Therefore, more research is needed to evaluate the safety and efficacy of these drugs combined with CDs in the treatment of COVID-19, as well as the long-term safety of all these associations.

In addition, some results indicate that CDs may also work as potential active pharmaceutical ingredients by themselves, which may influence the current regulatory landscape in using CDs. 

In the future, the use of computational approaches, namely molecular dynamics, may constitute an important tool to anticipate the solubility and interaction of some drugs with CDs [124,125,126]. 

## Figures and Tables

**Figure 1 ijms-24-02974-f001:**
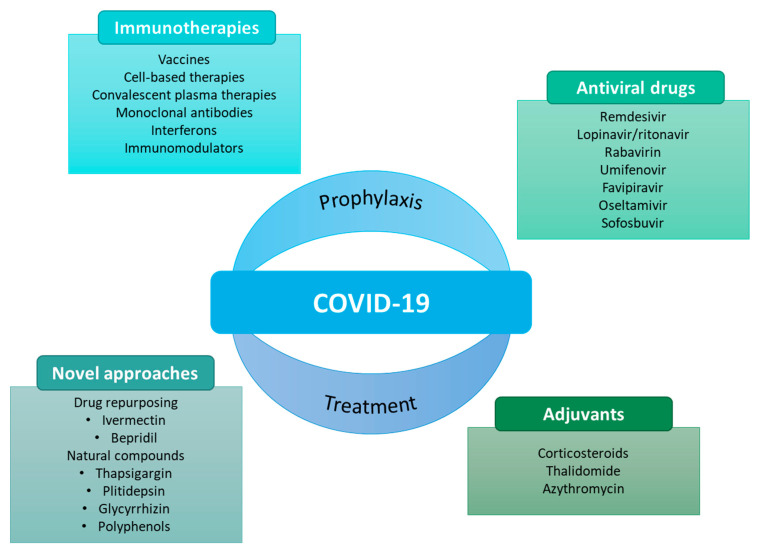
Some examples of prophylactic and therapeutic approaches against COVID-19 [12].

**Figure 2 ijms-24-02974-f002:**
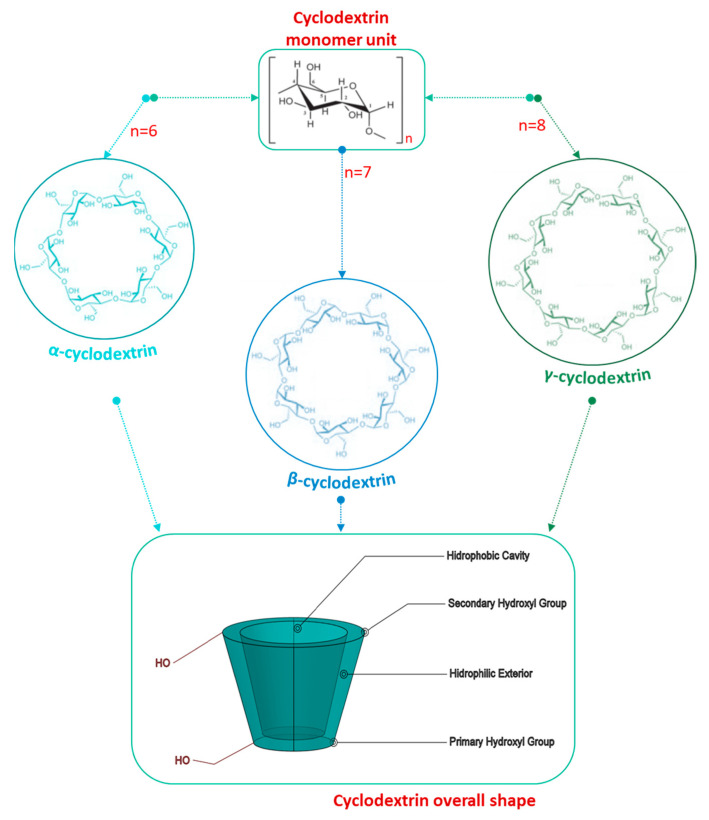
Illustration of cyclodextrins’ structure. Cyclodextrins are monomeric units of sugar molecules (glucopyranosides) arranged in rings. Depending on the number of sugar molecules (6, 7, or 8), they are classified as α-, β-, or γ-cyclodextrins, respectively, showing a general 3D overall shape [37].

**Figure 3 ijms-24-02974-f003:**
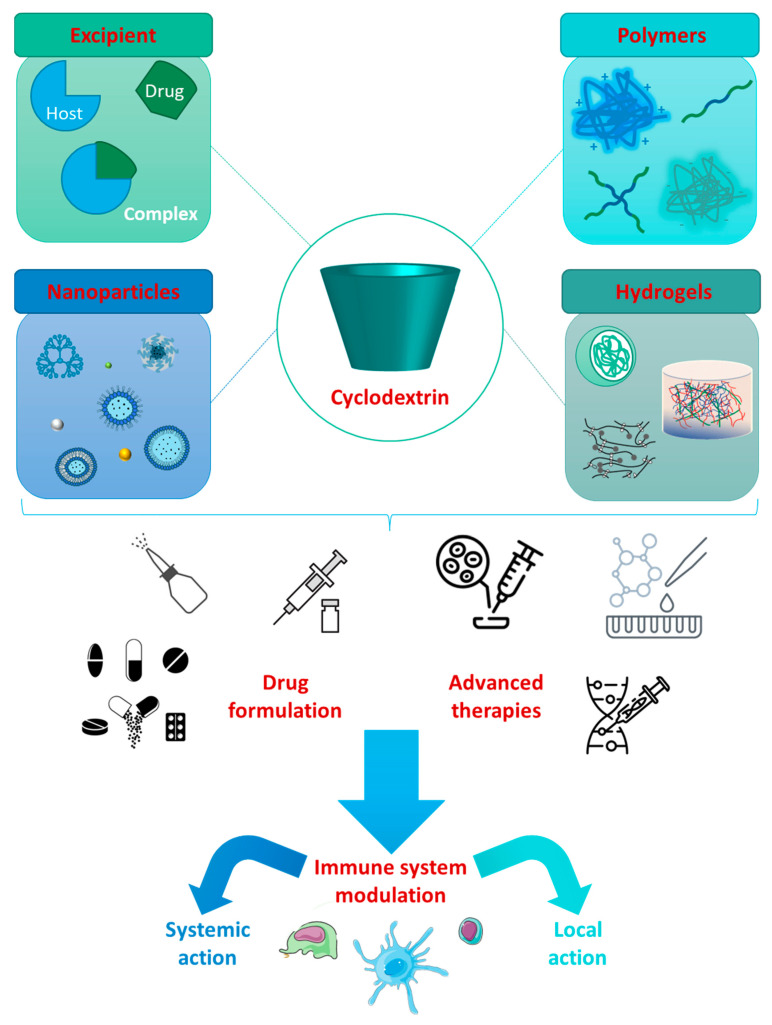
Cyclodextrins present versatile properties for formulating drug inclusion complexes and developing functional polymers, nanoparticles, and hydrogels. These cyclic oligosaccharides can promote the inclusion of therapeutic cargos, such as drugs, or advanced therapies, e.g., cells or genetic material, aiming for immunomodulatory outcomes by systemic or local action [38].

**Figure 4 ijms-24-02974-f004:**
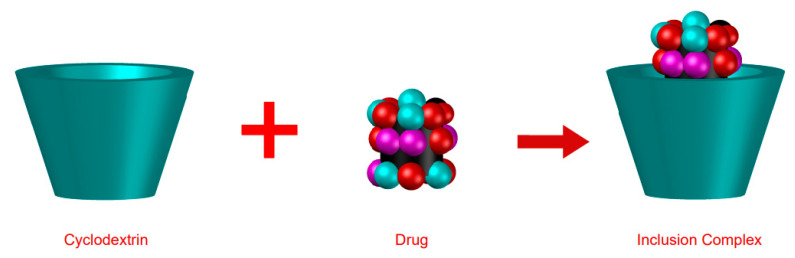
Formation of an inclusion complex.

**Figure 5 ijms-24-02974-f005:**
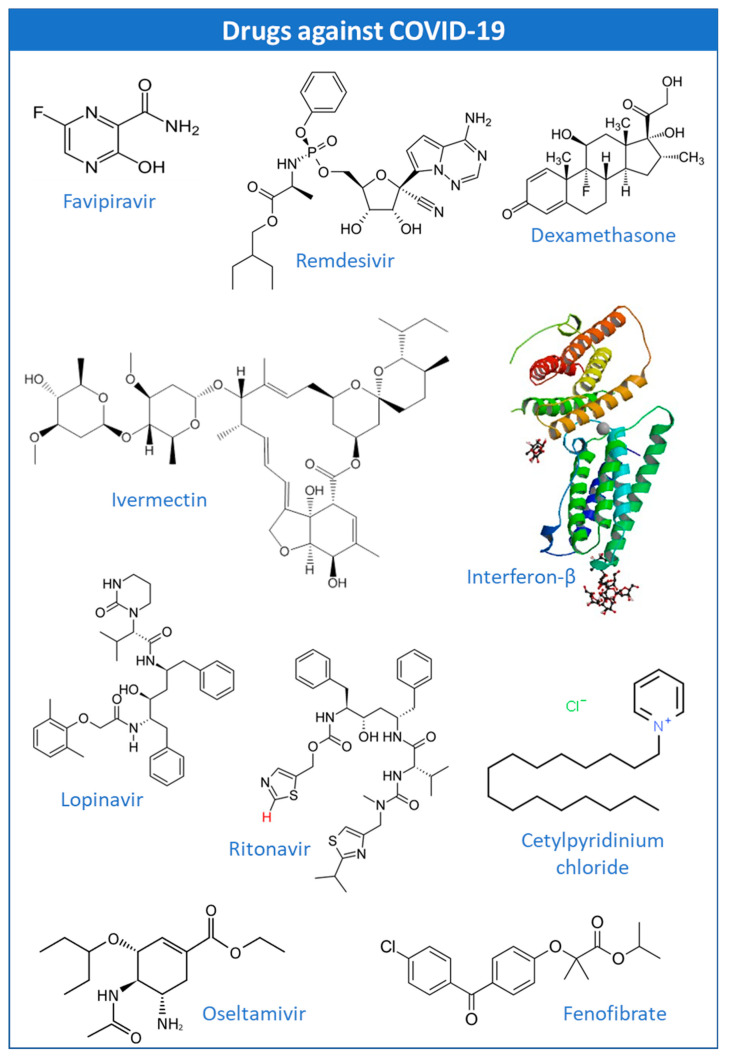
Structures of some drugs applied in clinical practice or clinical trials against COVID-19.

**Figure 6 ijms-24-02974-f006:**
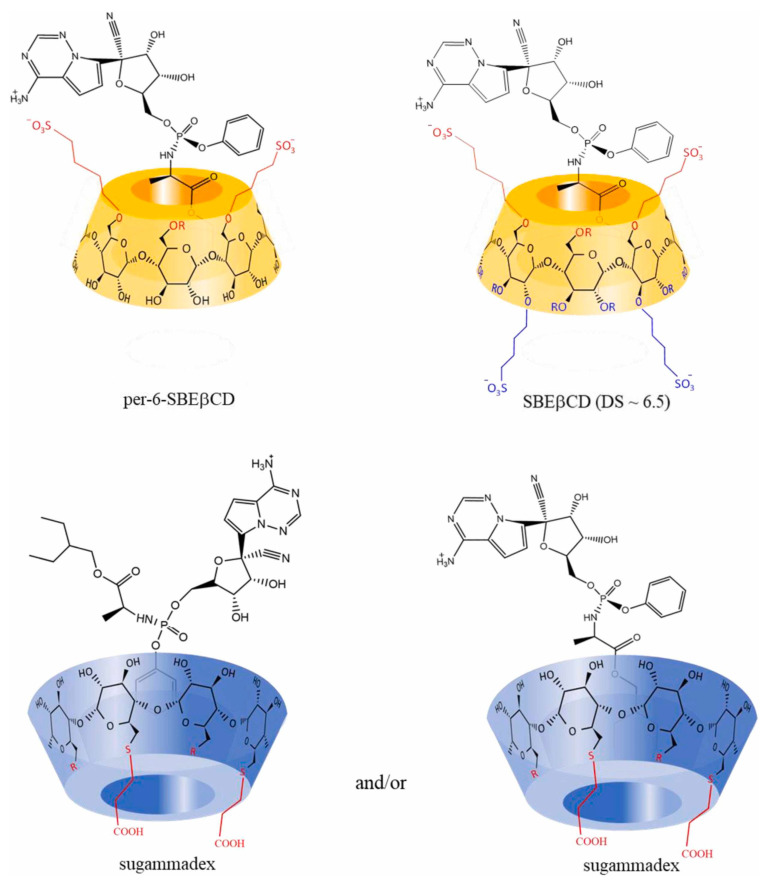
Suggested structures of inclusion complexes of the two β-cyclodextrins (CDs) derivatives (yellow) and remdesivir (REM) and the proposed sugammadex-REM complexes (blue) based on 2D ROESY NMR spectra (under pH 2.0 conditions). Reprinted from [46], Copyright (2022), with permission from Elsevier.

**Figure 7 ijms-24-02974-f007:**
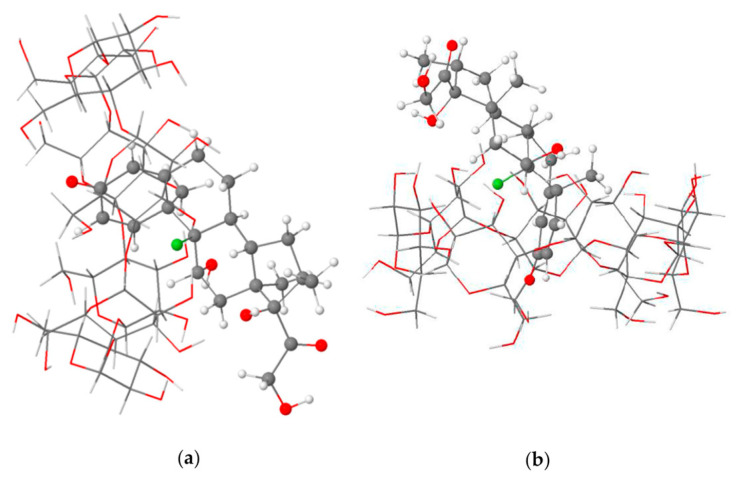
Side (**a**) and top (**b**) views of the partial inclusion of dexamethasone(DEX) in the β-CD cavity as calculated at the BLYP-D4/def2-TZVP level of theory in the gas phase. The structural analysis of the most stable DEX enclosed into the β-CD complex in the gas phase and an aqueous solution indicated the partial inclusion of the dexamethasone in the β-CD cavity from the wider rim (mode A). Adapted from [53], under Creative Common CC BY license.

**Figure 8 ijms-24-02974-f008:**
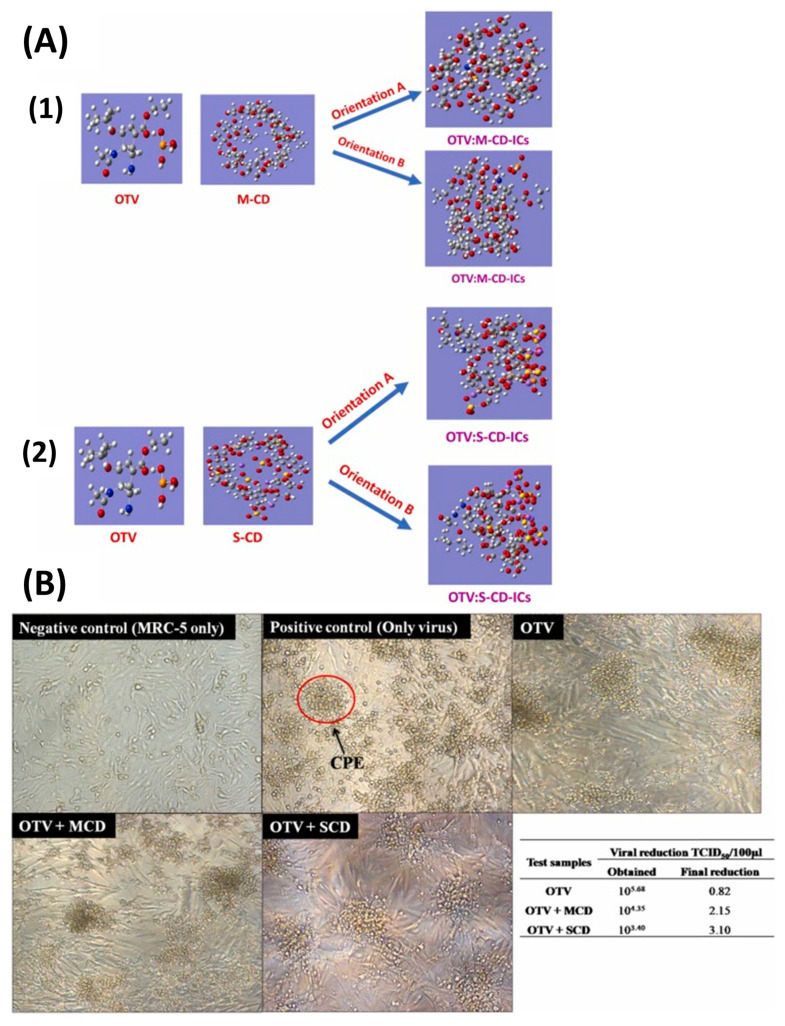
(**A**) Oseltamivir (OVT) orientation in (1) methylated-β-Cyclodextrin (M-CD) and (2) sulfated-β-Cyclodextrin (S-CD) inclusion complexes and (**B**) their antiviral effects in HCoV-229E infected MRC-5 cells. Briefly, OVT offers increased antiviral activity with more pronounced effects when using S-CD. Reprinted from [66], Copyright (2022), with permission from Elsevier.

**Figure 9 ijms-24-02974-f009:**
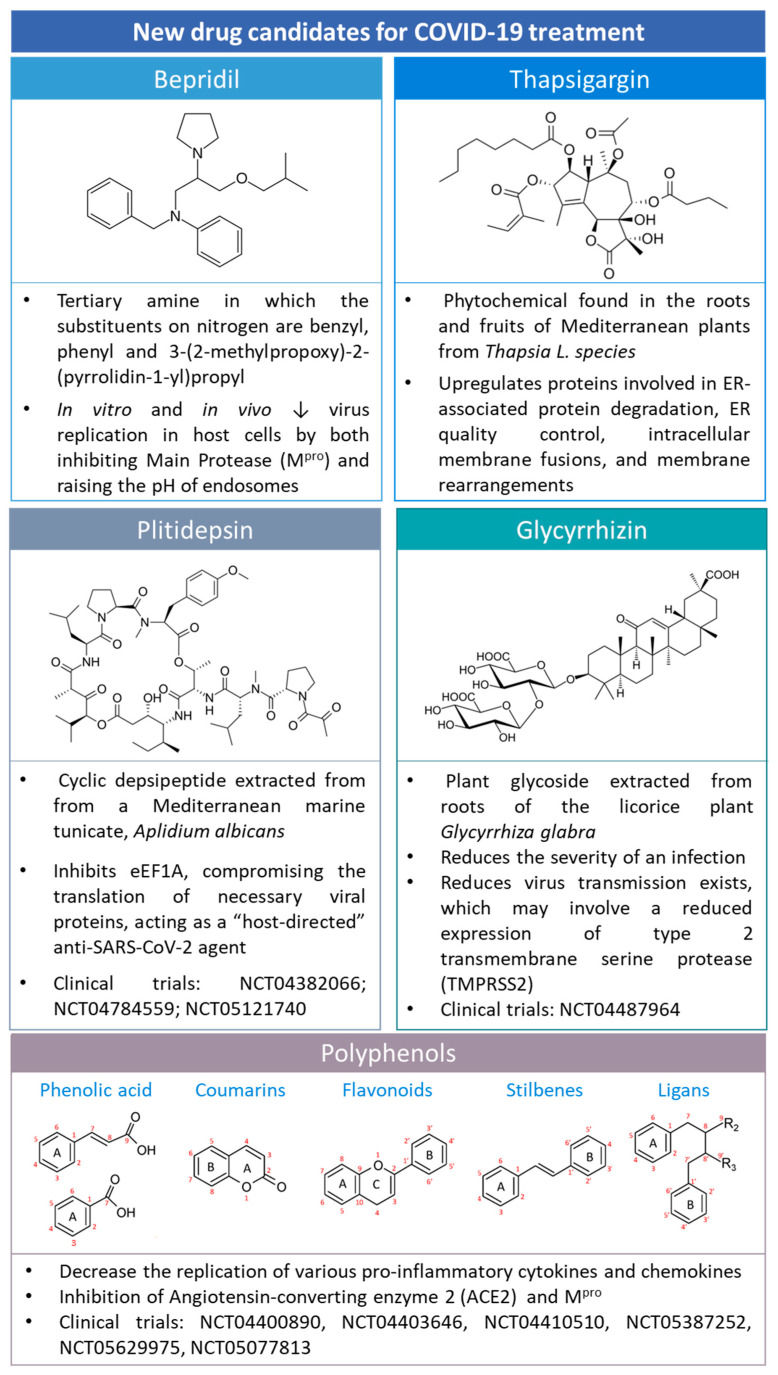
The chemical structure of drugs that are candidates for COVID-19 treatment, e.g., bepridil, thapsigargin, plitidepsin, glycyrrhizin, and polyphenols. Polyphenol structures were adapted from [75] under a Creative Common CC BY license.

**Figure 10 ijms-24-02974-f010:**
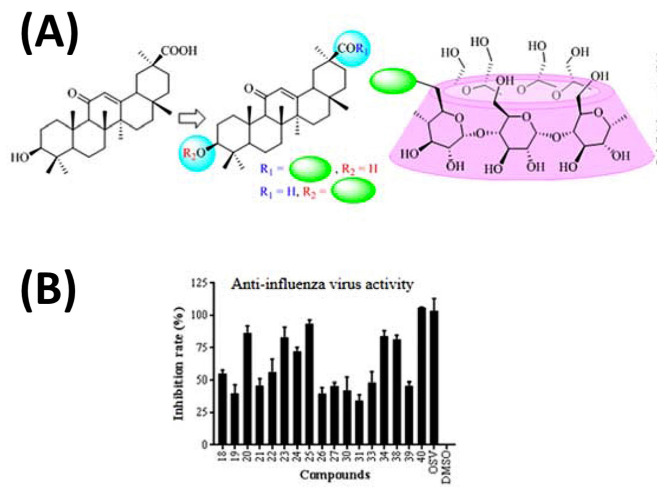
Schematic representation of the synthesis approach to conjugate glycyrrhetinic (GlyA) acid (**A**) to β-cyclodextrin and (**B**) their efficacy in inhibiting influenza virus infection. Accordingly, the conjugation of GlyA complexed to β-CD has demonstrated beneficial results for treating influenza virus agents. Adapted from [82], Copyright (2021), with permission from Elsevier.

**Table 1 ijms-24-02974-t001:** Potential therapeutic agents used in the treatment of COVID-19.

Drug	Drug Class	Mechanism of Action	Reference
Chloroquine and Hydroxychloroquine	Antimalarials	Increasing endosomal pH; interfering with the glycosylation of cellular receptors of SARS-CoV-2; immunomodulator.	[16]
Remdesivir	Antivirals	Interfering with the viral replication; inhibiting the viral RNA-dependent RNA polymerase (RdRp).	[7]
Favipiravir	Antivirals	Binds to the viral RdRp and reduces its reproduction.	[16]
Lopinavir and Rotinavir	Protease inhibitors	Could act by inhibiting SARS-CoV-2 protease for protein cleavage; interfering with virus replication.	[16]
Darunavir	Protease inhibitors	Could act by inhibiting SARS-CoV-2 protease for proteins cleavage; interfering with virus replication.	[16]
Niclosamide	Anthelmintics	Inhibiting replication and 3CL protease enzyme inhibition.	[14]
Ivermectin	Anthelmintics	Inhibits IMPα/β1 associated nuclear import of proteins of the virus.	[16]
Convalescent Plasma Therapy	Immunoglobulins	Non-neutralizing antibodies bind to the pathogen and contribute to prophylaxis and recovery.	[14]
Mesenchymal Stem Cell Therapy	Pluripotent stem cells	Prevent the release of cytokines.	[14]
Glycyrrhizin	Prenol lipids	Inhibits replication, adsorption, and penetration of the virus.	[14]
Cinanserin	5-HT 2A and 5-HT 2C receptor antagonist	Inhibition of the protease enzyme.	[14]
Dexamethasone	Corticosteroids	Reduces inflammation-induced lung damage and, consequently, inhibits the progression to respiratory failure.	[17]
IFN-β	Immunomodulators	Increases the production of anti-inflammatory cytokines and downregulates the production of pro-inflammatory cytokines.	[18]
Baricitinib	Janus kinase (JAK) inhibitors	Interfering with viral entry by inhibiting one of the endocytosis regulators and can prevent the activation of STAT.	[16]
Tocilizumab Bamlanivimab Etesevimab Lenzilumab Risankizumab CR3022	Monoclonal Antibodies	Neutralizing antibodies can block the entry of the virus into host cells and recruit host effector pathways to destroy virus-infected cells.	[7,16]
Camostat Mesylate	Transmembrane protease, serine 2 (TMPRSS2) inhibitor	Interfering with viral entry.	[16]

**Table 2 ijms-24-02974-t002:** Characteristics of α-CD, β-CD, and γ-CD [4].

Cyclodextrin	α-CD	β-CD	γ-CD
Height [nm]	0.78	0.78	0.78
Inner Diameter [nm]	0.47–0.52	0.60–0.65	0.75–0.83
Outer Diameter [nm]	1.46	1.54	1.75
Synonyms	Cyclomaltohexaose	Cycloheptaamylose	Cyclomaltooctaose
Molecular Formula	C_36_H_60_O_30_	[C_6_H_10_O_5_]_7_	C_48_H_80_O_40_
M_w_ [g/mol]	972	1132	1297
Solubility in Water at Room Temperature [mg/mL]	130	18.40	249
Hydrogen Bound Donor Count	18	21	24
Hydrogen Bond Acceptor Count	30	35	40

**Table 3 ijms-24-02974-t003:** Clinical trials using favipiravir for the treatment of COVID-19. Data was collected from clinicaltrials.gov on 28 November 2022. Inclusion criteria: Recruiting, adult population, from 1 January 2022 to 31 December.

NCT Number	Phase	Therapeutic Regimen		
		Dose	Administration Type	Other Drugs
NCT04613271	3	A total of 1600 mg twice a day on day 1 and 600 mg twice a day, on day 7–14	-	Favipiravir + Azithromycin
NCT04918927	2	A total of 800 mg twice daily on day 1, followed by 400 mg 4 times daily from day 2 to day 7	-	Favipiravir + Nitazoxanide
NCT05014373	3	A total of 1800 mg twice daily for one day, followed by 800 mg (4 tablets) twice daily	Oral	Favipiravir + Standard of Care
NCT04694612	3	A total of 1800 mg on D1 twice + 800 mg twice a day from day 2 for a total of 5 days	-	Remdesivir
NCT04718285	2	A total of 200 mg in a regimen of a 1600 mg twice daily loading dose followed by 600 mg twice daily	Oral	Montelukast + Favicovir (Favipiravir)
NCT05041907	2	A total of 1800 mg on D0 twice + 800 mg twice daily for a further 6/7 days	-	-

**Table 4 ijms-24-02974-t004:** Clinical trials using remdesivir for the treatment of COVID-19. Data was collected from clinicaltrials.gov on 28 November 2022. Inclusion criteria: Recruiting, adult population, from 1 January 2022 to 31 December 2022.

NCT Number	Phase	Therapeutic Regimen		
		Dose	Administration Type	Other Drugs
NCT04431453	2/3	-	IV	-
NCT04713176	3	200 mg	IV	DWJ1248 with Remdesivir Placebo with Remdesivir
NCT04738045	4	A loading dose of 200 mg then 100 mg once daily and lopinavir/ritonavir at a dose of 400/100 once daily for 5 days	IV	Lopinavir/ Ritonavir and Remdesivir
NCT04970719	3	A total of 200 mg followed by 100 mg once a day	IV	-
NCT04978259	4	-	IV	-
NCT04694612	3	A total of 200 mg followed by 100 mg daily	IV	-
NCT04779047	4	A total of 200 mg on day 1 then 100 mg once daily for 5 days	IV	-
NCT04693026	3	A total of 200 mg on day 1 then 100 mg daily	IV	Remdesivir + Baricitinib
NCT04321993	2	A total of 200 mg on day 1 then 100 mg daily	IV	Remdesivir (antiviral) + Barictinib
NCT04488081	2	A total of 200 mg on day 1 then 100 mg daily	IV	-

**Table 5 ijms-24-02974-t005:** Clinical trials using dexamethasone for the treatment of COVID-19. Data was collected from clinicaltrials.gov on 28 November 2022. Inclusion criteria: Recruiting, adult population, from 1 January 2022 to 31 December 2022.

NCT Number	Phase	Therapeutic Regimen		
		Dose	Administration Type	Other Drugs
NCT04663555	4	A total of 20 mg once daily followed by 10 mg once daily	IV	-
NCT05062681	4	8 mg 12 h	-	-
NCT04970719	3	6 mg	IV	Remdesivir + Dexamethasone
NCT04836780	3	A total of 6 mg once daily	-	-
NCT04452565	2/3	4 mg	-	NA-831+ Dexamethasone Atazanavir + Dexamethasone
NCT04528329	4	Early vs. late use	Early vs. late use	-
NCT04890626	3	-	-	Baricitinib + dexamethasone
NCT04826822	3	A total of 2 mg twice daily	Oral	Spironolactone + Dexamethasone
NCT05279391	-	A total of 6–8 mg once daily	-	-
NCT04784559	3	Country-specific product information	Country-specific product information	Plitidepsin + Dexamethasone
NCT04545242	4	A total of 6 mg/day or 20 mg daily	IV	-
NCT04488081	2	A total of 6 mg once daily	IV or oral	-

**Table 6 ijms-24-02974-t006:** Clinical trials using ivermectin for the treatment of COVID-19. Data was collected from clinicaltrials.gov on 28 November 2022. Inclusion criteria: Recruiting, adult population, from 1 January 2022 to 31 December 2022.

NCT Number	Phase	Therapeutic Regimen		
		Dose	Administration Type	Other Drugs
NCT05155527	2	A total of 600 mcg/kg once daily	Oral	Ivermectin + Favipiravir
NCT04681053	3	6 mg	Oral and inhaled	-
NCT05305560	2	A total of 200 mcg/kg on D1 then 100 mcg/kg daily from D2 to D28	Oral	-
NCT04723459	-	-	-	Ivermectin impregnated mask
NCT04729140	4	200 mcg/kg (3 mg)	Oral	Ivermectin + Doxycycline
NCT04834115	3	A 200 mcg/kg single dose, maximum dose of 18 mg	Oral	-
NCT04716569	2/3	-	Intranasal spray	-
NCT04472585	1/2	A total of 200 ug/kg body weight once every 48 h	Subcutaneous	Ivermectin with Zinc
		0.2 mg/kg/day	Oral	
NCT04959786	2/3	-	-	Ivermectin, ribavirin, nitazoxanide and zinc
NCT04435587	4	A total of 600 mcg/kg/day once daily for 3 days	Oral	-
NCT04779047	4	36 mg	-	Hydroxychloroquine, Tocilizumab, Ivermectin
NCT05045937	-	0.4 mg/kg	-	-
NCT04885530	3	7 mg tablets	Oral	-
NCT04951362	2/3	-	Intranasal spray	-
NCT04351347	2/3	Larger doses	-	-
NCT05041907	2	A total of 600 mcg kg/day for 7/7 days	-	-
NCT04703608	3	A total of 0.3–0.4 mg/Kg daily for 3 days	-	-
NCT02735707	3	A total of 0.2 mg/kg once daily with a maximum daily dose of 24 mg/day.	Enteral	-

**Table 7 ijms-24-02974-t007:** Clinical trials using lopinavir and ritonavir for the treatment of COVID-19. Data was collected from clinicaltrials.gov on 28 November 2022. Inclusion criteria: Recruiting, adult population, from 1 January 2022 to 31 December 2022.

NCT Number	Phase	Therapeutic Regimen		
		Dose	Administration Type	Other Drugs
NCT04738045	4	A Remdesivir 200 mg loading dose then 100 mg once daily and lopinavir/ritonavir at a dose of 400/100 once daily for 5 days	IV	Lopinavir/ Ritonavir and Remdesivir
NCT04779047	4	A dose of remdesivir 200 mg at day 1 then 100 mg once daily for 5 days and lopinavir/ritonavir at a dose of 400/100 once daily for 5 days, plus 800 mg of tocilizumab once	IV	Lopinavir/ Ritonavir, Remdesivir, Tocilizumab
NCT04403100	3	Hydroxychloroquine 400 mg: Loading dose of two tablets followed by one tablet of 400 mg on the following 9 days. Lopinavir/ ritonavir 200/50 mg: Loading dose of four tablets twice a day on day 1 followed by two tablets twice a day on the following 9 days	Oral	Hydroxychloroquine plus Lopinavir/ Ritonavir
NCT04466241	2/3	Lopinavir/ritonavir 200 mg/50 mg: two tablets morning and evening from day 1 to day 10. Telmisartan 40 mg: 1 tablet daily from day 1 to day 10	Oral	Lopinavir/ritonavir + telmisartan
		Lopinavir/ritonavir 200 mg/50 mg: two tablets morning and evening from day 1 to day 10. Atorvastatin 20 mg: 1 tablet daily from day 1 to day 10		Lopinavir/ritonavir + atorvastatin
NCT04351724	2/3	A dose of 200 mg/50 mg 4-0-4 on day 1 and 3-0-3 thereafter	-	-
NCT04381936	2/3	Lopinavir/ritonavir 400 mg/100 mg every 12 h for 10 days	By mouth (or nasogastric tube)	-
NCT04390152	1/2	Hydroxychloroquine 400 mg + lopinavir/ritonavir 400/100 or azithromycin 500 mg	-	Hydroxychloroquine, lopinavir/ritonavir and ventilation support plus placebo
NCT04380818	-	Ritonavir/lopinavir 400/100 mg/12 h for 7–10 days	-	-
NCT04410510	2/3	Lopinavir/ritonavir: 200/50 mg or 400/100 mg capsules every 12 h for 7 to 14 days. Hydroxychloroquine: 200 mg tab with a load of 400 mg every 12 h the first day, followed by 200 mg every 12 h for 10 days. P2Et active extract capsule equivalent to 250 mg of P2Et every 12 h for 14 days	-	Lopinavir/ritonavir, Hydroxychloroquine, P2Et active extract capsule
NCT02735707	3	Lopinavir/ritonavir-400/100 mg	Enteral	-
		or 5 mL 80/20 mg per mL solution suspension, every 12 h	Via gastric tube	

**Table 8 ijms-24-02974-t008:** Summary of clinical trials involving cyclodextrins for the treatment of COVID-19.

Trial ID	Date Registration	National Competent Authority	Active Substances	Pharmaceutical Form	Doses and Route of Administration	Trial Status
CTRI/2021/05/033744	24 May 2021	IN-CDSCO	Remdesivir (GS-5734)	Lyophilizate for solution for infusion	A total of 200 mg on day 1, followed by 100 mg for the next 4 days Intravenous	Ongoing
EUCTR2020-003486-19-GB	14 August 2020	UK - MHRA	Sulforadex or Sulforaphane/α-Cyclodextrin complex (SFX-01)	Hard Capsules	300 mg Oral	GB-no longer in EU/EEA
EUCTR2020-001803-17-GB	5 June 2020	UK–MHRA	Remdesivir (GS-5734)	Lyophilizate for solution for infusion	100 mg Intravenous	GB-no longer in EU/EEA
	10 August 2020	IT-Italian Medicines Agency				Ongoing

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
