# Peer review of "The Role of Cyclodextrins in COVID-19 Therapy—A Literature Review"

_ijms, 2023, doi:10.3390/ijms24032974_

Round 1

Reviewer 1 Report

Paper needs to be improved in following manners, this is a nice study however, following

questions are necessary to be answered before further processing

a. This would be beneficial if authors could provide more details at the end of introduction

specifically stating the objective of the paper, although this is explained but needs a little more

clarity.

b. Authors need to update the survey of literature for more recent papers specifically published in

the recent years 2022 etc.

The authors should elaborate on their new findings that are worthy of consideration for publication in a journal, below some proposed work:

https://doi.org/10.1016/j.enganabound.2022.11.033

https://doi.org/10.1016/j.enganabound.2022.10.034

https://doi.org/10.1142/S0217979223501473

https://doi.org/10.1016/j.advengsoft.2022.103267

https://doi.org/10.1016/j.seta.2022.102408

https://doi.org/10.1016/j.ijhydene.2022.07.140

https://doi.org/10.1016/j.jics.2022.100617

https://doi.org/10.1016/j.jobe.2022.104328

c. Language of the paper needs professional touch ups as there are typos and errors in some parts

of paper and they need to be reduced.

d. In the conclusion section, authors need to focus on the outcomes of their study with salient

findings only, keep them brief, as more explanation is already added in the results and

discussion section.

e. Results and discussion section is well explained, please try to look at figures in this section they might need more explanation if needed.

f. Altogether after these improvements are properly made, paper would be in a decent shape and

can be considered for publication if revised well.

g. Please give a bird eye view picture of your finding in abstract.

Author Response

Manuscript ID: ijms-2146498

Title: The role of Cyclodextrins in COVID-19 – a literature review

Dear Editor,

Doctor Bruno Zhang

The authors appreciate the careful review and greatly acknowledge the reviewer’ on our manuscript. We would like to kindly inform you that to help us in our revision, we have included another author, Inês Silva, with the approval of all authors. We have carefully revised the manuscript and have made the recommended changes and answered in detail to the questions raised. Additional information was also added when appropriate. All changes made to the text are highlighted in yellow colour and marked up using the “Track Changes” function.

# Reviewer 1

Paper needs to be improved in following manners, this is a nice study however, following questions are necessary to be answered before further processing

  1. This would be beneficial if authors could provide more details at the end of introduction specifically stating the objective of the paper, although this is explained but needs a little more clarity.

Response:

Thank you very much for your pertinent suggestion. A more detail description was provided accordingly, in order to clarify the objective of the paper. The text was modified as follows:

“The potential application of CDs in the encapsulation of antiviral drugs such as Favipiravir (FPV), Remdesivir (REM), Dexamethasone (DEX), Ivermectin (IVM), Hydroxy(chloroquine) (HCQ), Interferon-beta (IFN-β), Lopinavir-Ritonavir (LPV-RTV), Oseltamavir (OTV), and Fenofibrate has been explored with some promising features. This paper also brings together the ongoing clinical trials with these drugs, at the time of this review, to provide more detailed information about the main specifications of their use, and allow a clearer analysis of the feasibility of their incorporation into CDs. In addition, this work provides updated information about new proposals for therapeutic approaches for the treatment of COVID-19, highlighting new candidate drugs for this purpose Bepridil, Glycyrrhizin, Plitidepsin, Thapsigargin, and Polyphenols. Experimental studies of these drugs with CDs are also mentioned as well as the ongoing clinical trials.

This review also aims to raise awareness of the importance of toxicological analysis of CDs, focusing on aspects such as the daily dose administered, the route of administration, and the type of cyclodextrin. These aspects, together with the information collected from clinical and experimental studies, are essential to allow conclusions on the viability of incorporating these drugs into CDs.”

  1. Authors need to update the survey of literature for more recent papers specifically published in the recent years 2022 etc.

The authors should elaborate on their new findings that are worthy of consideration for publication in a journal, below some proposed work:

  1. https://doi.org/10.1016/j.enganabound.2022.11.033
  2. https://doi.org/10.1016/j.enganabound.2022.10.034
  3. https://doi.org/10.1142/S0217979223501473
  4. https://doi.org/10.1016/j.advengsoft.2022.103267
  5. https://doi.org/10.1016/j.seta.2022.102408
  6. https://doi.org/10.1016/j.ijhydene.2022.07.140
  7. https://doi.org/10.1016/j.jics.2022.100617
  8. https://doi.org/10.1016/j.jobe.2022.104328

Response:

The authors thank the reviewer’s comment. Accordingly, we have included the following information:

“Moreover, the use of computational approaches, namely molecular dynamics may constitute an important tool to anticipate the solubility and interaction of some drugs in CDs.[126–128]”

 in the section “Final remarks and future perspectives” to address the updates in molecular dynamics, mainly focused on its application for cyclodextrins.

Moreover, the following references were added:

  1. Boroushaki, T.; Dekamin, M.G. Interactions between β-cyclodextrin as a carrier for anti-cancer drug delivery: a molecular dynamics simulation study. J. Biomol. Struct. Dyn. 2022, doi:10.1080/07391102.2022.2164058.
  2. Cao, C.; Deng, C.; Hu, J.; Zhou, Y. Formation and molecular dynamics simulation of inclusion complex of large-ring cyclodextrin and 4-terpineol. J. Food Sci. 2022, 87, 4609–4621, doi:10.1111/1750-3841.16303.
  3. Raffaini, G.; Elli, S.; Ganazzoli, F.; Catauro, M. Inclusion Complexes Between β‐cyclodextrin and the Anticancer Drug 5‐Fluorouracil for its Solubilization: a Molecular Dynamics Study at Different Stoichiometries. Macromol. Symp. 2022, 404, 2100305, doi:10.1002/masy.202100305.

  1. Language of the paper needs professional touch ups as there are typos and errors in some parts of paper and they need to be reduced.

Response:

Thank you very much for your pertinent advice. We revised and corrected all the manuscript in this regard. All changes are highlighted in yellow throughout the text.

  1. In the conclusion section, authors need to focus on the outcomes of their study with saliente findings only, keep them brief, as more explanation is already added in the results and discussion section.

Response:

Thank you very much for your pertinent criticism. We appreciate and agree with your suggestion. This section was revised accordingly. The necessary modifications are highlighted and the manuscript was corrected as it follows:

“In addition, some results indicate that CDs may also work as potential active pharmaceutical ingredients by itself, which may influence the current regulatory landscape in using CDs.”

Moreover, the following reference was included:

  1. Di Cagno, M.P. The potential of cyclodextrins as novel active pharmaceutical ingredients: A short overview. Molecules 2017, 22.

  1. Results and discussion section is well explained, please try to look at figures in this section they might need more explanation if needed.

Response:

The authors appreciate the reviewer’s comment. As recommended, some figure captions were updated, particularly, Figure 6, 9, 10 and 11.

  1. Altogether after these improvements are properly made, paper would be in a decent shape and can be considered for publication if revised well.

Response:

Thank you very much for reviewing our document and for the constructive comments provided. We feel grateful that it may seem appropriate for publication after the corrections tackled. 

  1. Please give a bird eye view picture of your finding in abstract.

Response:

Thank you very much for the observation. We appreciate and agree with your suggestion. Accordingly, we added a brief information in the abstract. All the modifications are highlighted in the text. The manuscript was corrected as it follows:

“This review besides presenting studies on the inclusion of antiviral drugs in cyclodextrins, aims to summarize some currently available prophylactic and therapeutic schemes against COVID-19, highlighting those that already make use of cyclodextrins for their complexation. In addition, some new therapeutic approaches are underscored, and the potential application of cyclodextrins to increase their promising application against COVID-19 will be addressed. This review describes the instances in which the use of cyclodextrins promotes increased bioavailability, anti-viral action, and solubility of the drugs under analysis. The potential use of CDs as an active ingredient is also covered.”

Reviewer 2 Report

1.       Please update the title to “The role of cyclodextrin in COVID-19 therapy” because CDs by themselves have no therapeutic effect.  They enhance the solubility of therapeutic drugs.

2.       Could you please put line numbers from page number 6?

3.       Section 4.2.10 for CPC in the last sentence of the document mentions about CA patent: CA1314225C. It is an expired patent.  Please include this information.

4.       Section 5.3 “Plitidepsin” could you please define Eukaryotic translation elongation factor 1 alpha in the first referred under reference 83. Second paragraph defines the entire nomenclature.

5.       Please include use of hydroxypropyl beta cyclodextrin indicated for the development of oral medicine for COVID reference: PMID: 34726479. Thank you.

Author Response

Manuscript ID: ijms-2146498

Title: The role of Cyclodextrins in COVID-19 – a literature review

Dear Editor,

Doctor Bruno Zhang

The authors appreciate the careful review and greatly acknowledge the comments that the Reviewer has provided on our manuscript. We would like to kindly inform you that to help us in our revision, we have included another author, Inês Silva, with the approval of all authors. We have carefully revised the manuscript and have made the recommended changes and answered in detail to the questions raised. Additional information was also added when appropriate. All changes made to the text are highlighted in green colour and marked up using the “Track Changes” function.

# Reviewer 2

  1. Please update the title to “The role of cyclodextrin in COVID-19 therapy” because CDs themselves have no therapeutic effect. They enhance the solubility of therapeutic drugs.

Response:

The authors acknowledge the Reviewer’s comment. Accordingly, the title has been updated to: “The role of cyclodextrin in COVID-19 therapy – a literature review.”

  1. Could you please put line numbers from page number 6?

Response:

The authors thank the Reviewer’s comment. Consequently, the line numbers have been included.

  1. Section 4.2.10 for CPC in the last sentence of the document mentions about CA patent: CA1314225C. It is an expired patent. Please include this information.

Response:

The authors appreciate the Reviewer’s comment. Therefore, this information has been included as follows:

“However, an expired patent (CA1314225C, Canada) mentions the application of CDs to inclose CPC.”.

  1. Section 5.3 “Plitidepsin” could you please define Eukaryotic translation elongation factor 1 alpha in the first referred under reference 83. Second paragraph defines the entire nomenclature.

Response:

The authors acknowledge the Reviewer’s comment. Accordingly, this information has been updated.

  1. Please include use of hydroxypropyl beta cyclodextrin indicated for the development of oral medicine for COVID reference: PMID: 34726479. Thank you.

Response:

The authors would like to thank the Reviewer’s comment. Appropriately, the information has been included as follows:

“Moreover, the combination of ritonavir with other antiviral agents, such as nirmatrevil (PF-07321332), has been addressed in clinical trials and has shown clinical efficacy in reducing hospitalization by 80%.[40] PF-07321332 is a reversible covalent inhibitor of the Mpro related to SARS-CoV-2 that binds to the catalytic cysteine (Cys145), interrupting the viral replication cycle.[41,42] According to EMA/783153/2021 (16 December 2021), PF-07321332 belongs to the BCS II/IV, presenting low solubility with permeability to be clarified. The use of CDs for oral administration is not fully exploited. However, the application of 2-hydroxypropyl-β- CD in the formulation of PF-07321332 for IV administration in monkeys has been tested. [41]”

For this, three references have been included, namely the one suggested by the Reviewer (PMID: 34726479, Ref. 41):

  1. Chaves, O.A.; Sacramento, C.Q.; Ferreira, A.C.; Mattos, M.; Fintelman-Rodrigues, N.; Temerozo, J.R.; Vazquez, L.; Pinto, D.P.; da Silveira, G.P.E.; da Fonseca, L.B.; et al. Atazanavir Is a Competitive Inhibitor of SARS-CoV-2 Mpro, Impairing Variants Replication In Vitro and In Vivo. Pharmaceuticals 2021, 15, 21, doi:10.3390/ph15010021.
  2. Owen, D.R.; Allerton, C.M.N.; Anderson, A.S.; Aschenbrenner, L.; Avery, M.; Berritt, S.; Boras, B.; Cardin, R.D.; Carlo, A.; Coffman, K.J.; et al. An oral SARS-CoV-2 Mpro inhibitor clinical candidate for the treatment of COVID-19. Science (80-. ). 2021, 374, 1586–1593, doi:10.1126/science.abl4784.
  3. Hau, R.K.; Wright, S.H.; Cherrington, N.J. PF-07321332 (Nirmatrelvir) does not interact with human ENT1 or ENT2: Implications for COVID-19 patients. Clin. Transl. Sci. 2022, 15, 1599–1605, doi:10.1111/cts.13292.
